

# Genome-wide association study of pigmentary traits (skin and iris color) in individuals of East Asian ancestry

Lida Rawofi[1], Melissa Edwards[1], S Krithika[1], Phuong Le[1], David Cha[1], Zhaohui Yang[2], Yanyun Ma[3], Jiucun Wang[4], Bing Su[5], Li Jin[4], Heather L. Norton[6] and Esteban J. Parra[1]

[1] Department of Anthropology, University of Toronto at Mississauga, Mississauga, Canada
[2] Yunnan Key Laboratory of Primate Biomedical Research, Institute of Primate Translational Medicine, Kunming University of Science and Technology, Kunming, China
[3] MOE Key Laboratory of Contemporary Anthropology, School of Life Sciences, Fudan University, Shanghai, China
[4] State Key laboratory of Genetic Engineering, Collaborative Innovation Center for Genetics and Development, School of Life Sciences, Fudan University, Shanghai, China
[5] State Key Laboratory of Genetic Resources and Evolution, Kumming Institute of Zoology, Chinese Academy of Sciences, Kunming, China
[6] Department of Anthropology, University of Cincinnati, Cincinnati, United States of America

Corresponding author
Esteban J. Parra,
esteban.parra@utoronto.ca

## ABSTRACT

**Background**. Currently, there is limited knowledge about the genetics underlying pigmentary traits in East Asian populations. Here, we report the results of the first genome-wide association study of pigmentary traits (skin and iris color) in individuals of East Asian ancestry.

**Methods**. We obtained quantitative skin pigmentation measures (M-index) in the inner upper arm of the participants using a portable reflectometer ($N = 305$). Quantitative measures of iris color (expressed as L\*, a\* and b\* CIELab coordinates) were extracted from high-resolution iris pictures ($N = 342$). We also measured the color differences between the pupillary and ciliary regions of the iris (e.g., iris heterochromia). DNA samples were genotyped with Illumina's Infinium Multi-Ethnic Global Array (MEGA) and imputed using the 1000 Genomes Phase 3 samples as reference haplotypes.

**Results**. For skin pigmentation, we did not observe any genome-wide significant signal. We followed-up in three independent Chinese samples the lead SNPs of five regions showing multiple common markers (minor allele frequency $\geq$ 5%) with good imputation scores and suggestive evidence of association ($p$-values $< 10^{-5}$). One of these markers, rs2373391, which is located in an intron of the ZNF804B gene on chromosome 7, was replicated in one of the Chinese samples ($p = 0.003$). For iris color, we observed genome-wide signals in the OCA2 region on chromosome 15. This signal is driven by the non-synonymous rs1800414 variant, which explains 11.9%, 10.4% and 6% of the variation observed in the b\*, a\* and L\* coordinates in our sample, respectively. However, the OCA2 region was not associated with iris heterochromia.

**Discussion**. Additional genome-wide association studies in East Asian samples will be necessary to further disentangle the genetic architecture of pigmentary traits in East Asian populations.

## INTRODUCTION

Human pigmentation diversity is primarily driven by the type, amount and distribution of melanin in the skin, hair and iris. Pigmentation is a polygenic trait, and the last decade has witnessed numerous efforts to elucidate the genetic architecture of pigmentation through association and functional studies (*Lamason et al., 2005*; *Kayser et al., 2008*; *Visser, Kayser & Palstra, 2012*; *Liu et al., 2015*). There have also been important advances in the development of methods based on reflectance and bioimaging technologies, which have made it possible to obtain quantitative measurements of skin, hair and iris pigmentation (*Liu et al., 2010*; *Walsh et al., 2011*; *Edwards et al., 2012*; *Beleza et al., 2013a*; *Andersen et al., 2013*; *Norton et al., 2016*; *Edwards et al., 2016*; *Wollstein et al., 2017*). As a result of these efforts, dozens of genetic markers have been associated with pigmentary phenotypes. However, the overwhelming majority of studies have focused on European populations, and there are still substantial gaps in our understanding of the genetic basis of pigmentation in other population groups.

Nonetheless, available research has shown skin lightening in Europe and East Asia to have occurred independently through convergent evolution. In Europe, markers within the *HERC2* gene (e.g., rs12913832), which is close to *OCA2*, are strongly associated with blue eyes, and have also been associated with light skin pigmentation in GWAS. (*Kayser et al., 2008*; *Liu et al., 2015*; *Sturm et al., 2008*; *Visser, Kayser & Palstra, 2012*). The haplotype defined by rs12913832 is primarily restricted to Europe. In East Asia however, two non-synonymous variants in the *OCA2* gene; rs1800414 (His615Arg) and rs74653330 (Ala481Thr) have been associated with light skin and eye color (*Edwards et al., 2010*; *Abe et al., 2013*; *Eaton et al., 2015*; *Edwards et al., 2016*; *Norton et al., 2016*; *Yang et al., 2016*). Both of these polymorphisms are predicted to have a deleterious effect on the protein (*Eaton et al., 2015*). Functional studies have recently confirmed the role of rs1800414 in East Asian pigmentation (*Yang et al., 2016*). The haplotypes defined by rs1800414 and rs74653330 are restricted to East Asia but they have a very different geographic distribution. The SNP rs1800414 is very frequent across East Asia while rs74653330 is primarily restricted to the Altaic speaking populations from Northern East Asia and Mongolia (*Murray, Norton & Parra, 2015*). There is general agreement that these two non-synonymous mutations arose long after the split of European and East Asian populations (*Chen, Hey & Slatkin, 2015*; *Murray, Norton & Parra, 2015*; *Yang et al., 2016*). In addition to variants in the *OCA2* gene, a non-synonymous mutation at rs885479 (Arg163Gln) in *MC1R* has also been associated with pigmentation in East Asia (*Yamaguchi et al., 2012*). Interestingly, the derived 163Gln allele is present at very high frequencies in East Asian populations (>60%), but very low frequencies in European and African populations.

Most of the studies conducted in East Asian populations have been candidate gene studies. Here, we report the results of the first genome wide association study of pigmentary traits in an East Asian population sample. Skin pigmentation was measured with a portable

reflectometer while eye color was measured quantitatively (CIELab color space) from high-resolution pictures of the iris. The samples were genotyped with Illumina's MEGA array, and imputed with the 1KG Phase 3 reference samples. We followed up the main signals observed in our GWAS in two independent East Asian samples. This study provides important insights about the architecture of skin pigmentation in East Asian populations.

## MATERIALS AND METHODS

### Sample collection

Recruitment of study participants was carried out at the University of Toronto (Canada). All participants ranged between 18 and 35 yr of age and were recruited using online and print advertisements directed towards the University of Toronto student community. A personal questionnaire was administered to each participant to determine their age, sex and whether or not they had been diagnosed with any pigmentation-related diseases or disorders.

Biogeographical ancestry was determined using information from the personal questionnaire, which inquired about the ancestry, place of birth and first language of each participant's maternal and paternal grandparents. The sample used in this study comprised individuals who had grandparents from China, Japan, South Korea or Taiwan. In the few cases where information about grandparents was not available, we used information about both parents to assess biogeographical ancestry. The total number of individuals included in the study was 425. The study was approved by the University of Toronto Research and Ethics Board (Protocol Reference #27015), and all participants provided written informed consent. A 2-ml saliva sample was obtained from each participant using the Oragene-DNA (OG-500) collection kit (DNA Genotek, Ottawa, Canada). All participants were instructed not to eat, drink or smoke for at least 30 min prior to obtaining the sample to ensure maximal sample purity. DNA was isolated from each sample using the protocol provided by DNA Genotek and eluted in 500 ml of TE (10 mM Tris–HCl, 1 mM EDTA, pH 8.0) buffer. Prior to genotyping, 23 DNA samples were excluded due to poor DNA quality. The final number of samples genotyped was 402.

### Measurement of pigmentary traits

Skin pigmentation was measured quantitatively using the DSMII Dermaspectrometer (Cortex Technologies, Hadsund, Denmark), and melanin levels were reported as M index (individuals with higher $M$-values have darker pigmentation). Measurements were taken three times on the inner skin of the upper right arm and pigmentation was reported as the mean of the three measures, after excluding outliers. High-resolution pictures of the right iris of each participant were taken with a Fujifilm Finepix S3 Pro 12-megapixel DSLR mounted on a Nikor 105-mm macro lens. To control for lighting and exposure, photographs were taken with a coaxial biometric illuminator to deliver a constant and uniform source of light to each iris at 5,500 K (D55 illuminant). All photographs were taken under the same setting (*Delaneau, Marchini & Zagury, 2012*) conditions, with an aperture of f/19, exposure sensitivity (ISO) set at 200 and a shutter speed of 1/125 s. Iris pigmentation was digitally scored using a custom program designed to crop out both the

pupil and sclera to retain only the iris. A wedge of the iris was then extracted, and color scores in CIELab coordinates were calculated from the pupillary and ciliary zones. In addition to the L\*, a\* and b\* coordinates for the iris wedge, the program calculated the parameter delta, which describes color differences in the pupillary and ciliary regions of the iris. Detailed information about this program has been described in *Edwards et al. (2016)*.

## Genotyping, phasing and imputation

Genotyping was carried out with Illumina's Infinium Multi-Ethnic Global Array (MEGA) at the Clinical Genomics Centre (Mount Sinai Hospital, Toronto, Ontario, Canada) using standard protocols. The MEGA array, which includes approximately 1.7 million markers, was designed to capture common genome variation in diverse population groups. Four samples were included as blind duplicates, and the concordance rate was in all samples higher than 99.99%. We used the program GenomeStudio to carry out the basic QC steps recommended by Illumina. After this initial QC step, approximately 1.4 million were retained for further analyses. The number of autosomal markers included was approximately 1.36 million. We performed additional QC steps to remove samples and markers, according to the following criteria, Sample QC: 1/removal of samples with missing call rates <0.9, 2/removal of samples that were outliers in Principal Component Analysis (PCA) plots, 3/removal of samples with sex discrepancies, 4/removal of samples that were outliers for heterozygosity, and 5/removal of related individuals (pi-hat > 0.2). Marker QC: 1/removal of markers with genotype call rate <0.95, 2/removal of markers with Hardy-Weinberg $p$-values $<10^{-6}$, 3/removal of Insertion/Deletion (Indel) markers, 4/removal of markers with allele frequencies <0.01, 5/removal of markers not present in the 1000 Genomes reference panel, or that do not match on chromosome, position and alleles, 6/removal of A/T or G/C SNPs with MAF >40% in the 1000 Genomes East Asian reference samples, and 7/removal of SNPs with allele frequency differences >20% between the study sample and the 1000 Genomes East Asian reference sample. After these QC steps, we retained 377 samples and 520,076 markers.

After performing the QC steps described above, the samples were phased using the program SHAPEIT2 and imputed at the Sanger Imputation Service, using the Positional Burrows-Wheeler Transform (PBWT) algorithm (*Durbin, 2014*), and the samples of the 1000 Genomes as reference haplotypes.

## Population structure

We used the program EIGENSOFT to perform PCA and evaluate population stratification after pruning markers in high LD and removing regions showing high LD or genomic complexity.

## Statistical analyses

As a first step of the statistical analyses, we carried out a linear regression with $M$-values as the dependent variable, and sex and the first four Principal Component Axis as independent variables and saved the standardized residuals. A similar process was carried out for the L\*, a\*, b\* and delta iris values, but in this case, due to deviations from normality, the unstandardized residuals were transformed using the rank-based inverse normal

transformation. The $M$-value residuals and the L$^\star$, a$^\star$, b$^\star$ and delta transformed residuals were used as input for the association tests with the program SNPTEST v2 (*Marchini & Howie, 2010*), using an additive model and the expected test (e.g., using genotype dosages) in order to control for genotype uncertainty. For the L$^\star$, a$^\star$, b$^\star$ coordinates that define iris color, we also run a Bayesian Multiple Phenotype test implemented in the program SNPTEST (-mpheno option). This test evaluates the three coordinates jointly and provides a $\log^{10}$ Bayes Factor reporting the ratio of two probabilities: the probability of the data under an unconstrained model ($M_1$), and the probability of the data under a null model ($M_0$) in which there is no effect. For example, a $\log^{10}$ Bayes Factor of 3 indicates that the probability of the data under the model M1 is 1,000-fold higher than the probability of the data under the null model with no genotype effects.

Of the 377 samples that were retained after the post-genotyping QC step, some samples had missing phenotype data. The final number of samples with valid skin pigmentation data was 305, and the final number of samples with valid iris color data was 342.

## Annotation of genome-wide significant and suggestive signals

The genome-wide significant ($p < 10^{-8}$) and suggestive signals ($p < 10^{-5}$) identified in the statistical analyses were annotated using the online SNP-Nexus tool (http://snp-nexus.org/), which provides extensive annotations, including potential effects of non-synonymous coding SNPs on protein function (e.g., SIFT and Polyphen), potential regulatory effects (e.g., conserved transcription factor binding sites, microRNAs, Enhancers and CpG islands), evidence of evolutionary conservation (e.g., PHAST and GERP++) and evidence of association with complex diseases and disorders (e.g., GAD and NHGRI Catalogue of Published Genome-Wide Association studies).

## Replication in independent East Asian samples

The markers showing the strongest evidence of association with skin pigmentation were followed up in three independent Chinese samples, for which skin pigmentation measures (M-index or L$^\star$ values) were also available. The first two samples were collected by the Laboratory of Contemporary Anthropology at Fudan University between 2013 and 2015. Constitutive pigmentation was measured with a DSMII colormeter (Cortex Technology, Hadsund, Denmark) and pigmentation levels were reported as M-index values (darker pigmentation corresponds to higher M-index values). Pigmentation was estimated as the mean of three measures, after excluding outliers. The first sample was a cohort from the Jinan Military Hospital ($N = 559$). All the participants were male, and the mean age of this sample was 21.13 years. Constitutive pigmentation was measured in the buttocks during the month of September. The second sample was a cohort from Taizhou ($N = 568$), comprising 404 females and 164 males, with an average age of 44.32 years. Constitutive pigmentation was measured in the upper inner arm in the month of April. Genotyping of the SNPs was carried out using SNaPshot. In both samples, the association of the relevant markers with M-index values was carried out using linear regression, including age, rs1800414 and sex (only for the Taizhou cohort) as covariates. The third sample comprises 346 Han Chinese individuals (College students) for whom constitutive pigmentation was measured in the

buttocks and the upper inner arm. The study took place in the month of September. In this case, pigmentation was estimated as the mean of three measurements and reported using L* values (darker pigmentation corresponds to lower L* values). Genotyping was done using Sanger sequencing. The association of the variants with pigmentation was tested with the program PLINK, using sex and age as covariates. Unfortunately, to our knowledge, there have been no studies in East Asia reporting quantitative iris color estimates using the CIELab color space, so it was not possible to follow-up the signals identified for iris color in our East Asian samples.

## RESULTS

### Distribution of pigmentary traits in East Asian sample

Figure S1 shows the distribution of skin pigmentation values, expressed as the M-index. The average $M$-value was 37.83 (SD = 2.843). Figures S2A–S2C show the scatterplots of a* and b*, L* and a* and L* and b* iris color coordinates, respectively. The CIELab plots show that there is a substantial amount of variation in the three iris color coordinates, emphasizing the importance to use quantitative estimates of iris color, instead of categorical classifications (which in the case of this sample, would be restricted to "brown" color). Finally, the distribution of the delta values (difference in color coordinates between the pupillary and ciliary regions of the iris) are depicted in Fig. S3.

### Population structure

Figure S4 shows the representation of the first two axes of a PCA analysis of the East Asian sample included in this study, as well as the East Asian samples of the 1000 Genomes project (CDX: Chinese Dai in Xishuangbanna, KHV: Kinh in Ho Chi Minh City, Vietnam, CHB: Han Chinese in Beijing, CHS: Han Chinese South and JPT: Japanese in Tokyo). Most of the East Asian individuals included in this study overlap with the two Han Chinese samples (CHB and CHS) and a few individuals with the Japanese samples. Interestingly, the East Asian individuals who reported ancestry from Korea in our sample form a discrete cluster that is located between the Han Chinese and the Japanese 1000 Genomes samples.

### Results of genome-wide association study

Figure 1 shows a Manhattan plot reporting the association results for skin pigmentation. We did not observe any genome-wide significant signal ($p < 5 \times 10^{-8}$). Figure S5 shows the QQ plot corresponding to skin pigmentation. There was no evidence of genomic inflation (lambda: 1.00). Table 1 reports the list of the common markers (maf > 5%) with the lowest $p$-values identified in the GWA study. All the markers listed in the Table have good imputation info scores (info > 0.8), and correspond to regions with multiple significant markers (File S1). The regional plots for these regions are depicted in Fig. S6. Figure 2 shows the Manhattan plots reporting the association results for iris color (L*, a*, b* and delta). The QQ plots for these traits are depicted in Fig. S7. Again, there is no evidence of genomic inflation for any of the traits. We observed a genome-wide signal in the well-known *OCA2* region. In this region, the top signal was rs76930569, which had $p$-values of $9.27 \times 10^{-11}$ and $5.30 \times 10^{-12}$ for the a* and b* color dimensions, respectively.
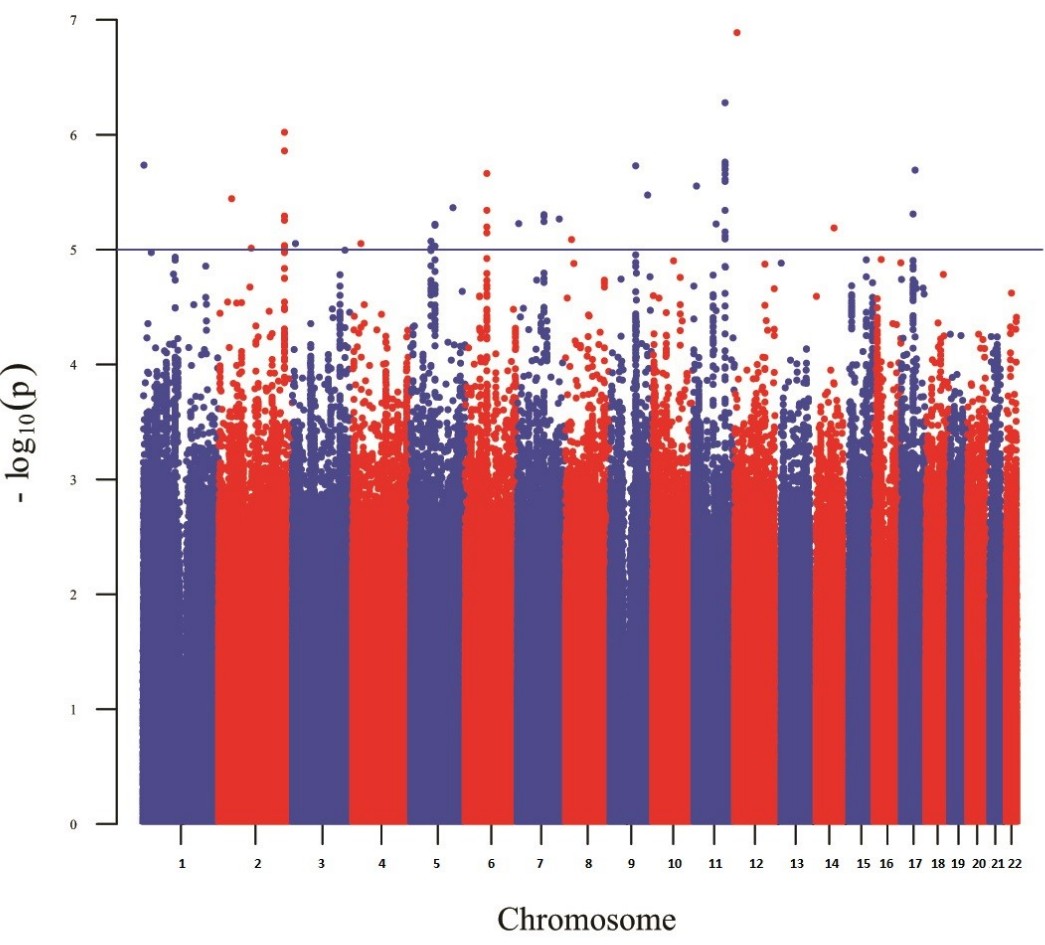

**Figure 1   Manhattan plot reporting the association results for skin pigmentation.**

The Bayesian multiple phenotype test implemented in SNPTEST, based on the L*, a* and b* dimensions, also provided strong support for the *OCA2* region, with a $\log^{10}$ Bayes Factor of 7. In addition to the *OCA2* region, we identified numerous regions showing multiple markers with suggestive associations ($p < 10^{-5}$) and good imputation scores. The lead signals for these regions are listed on Table 2, for each of the traits analyzed (L*, a*, b* and delta). In order to facilitate interpretation, we provide in the Table the *P*-values for all the traits. It can be seen that all the regions show nominal significance in more than one iris color dimension. Additional information about relevant markers within each region is provided in the File S1. The regional plots are depicted in Figs. S8–S11, respectively.

### Replication of suggestive skin pigmentation signals

The lead SNPs showing suggestive evidence of association ($p < 10^{-5}$) were followed up in three independent Chinese samples, for which skin pigmentation measures in the buttocks or inner arm were available (M-index or L* values). The results of the replication effort are shown in Table 3. One of the markers, rs2373391 located on the *ZNF804B* gene on chromosome 7, was replicated in the Physical Examination cohort. In this sample, the A

Rawofi et al. (2017), *PeerJ*, DOI 10.7717/peerj.3951

**Table 1** **Main signals observed in the GWAS of skin pigmentation (*M*-values).** We report the beta, SE and *P*-values obtained with the model using the standardized and unstandardized residuals.

| SNP | CHR | POS | NEA/EA | Gene | Frq EA | INFO | Beta[a] | SE[a] | P[a] | Beta[b] | SE[b] | P[b] |
|---|---|---|---|---|---|---|---|---|---|---|---|---|
| rs2003589 | 2 | 217527465 | T/C | *IGFBP2* | 0.063 | 0.843 | 0.885 | 0.177 | 9.54E–07 | 2.464 | 0.494 | 1.05E–06 |
| rs853807 | 5 | 67752638 | C/T | | 0.485 | 0.956 | −0.347 | 0.077 | 8.45E–06 | −0.961 | 0.213 | 9.75E–06 |
| rs57836066 | 6 | 71304950 | T/G | *RP11-134K13.4* | 0.086 | 0.979 | −0.661 | 0.137 | 2.17E–06 | −1.838 | 0.382 | 2.40E–06 |
| rs2373391 | 7 | 88449300 | T/A | *ZNF804B* | 0.681 | 0.987 | −0.384 | 0.083 | 4.98E–06 | −1.076 | 0.232 | 5.09E–06 |
| rs7945369 | 11 | 103425586 | T/C | | 0.537 | 0.968 | −0.405 | 0.079 | 5.26E–07 | −1.159 | 0.224 | 3.97E–07 |

**Notes.**
[a] Beta estimate using standardized residuals.
[b] Beta estimate using unstandardized residuals.

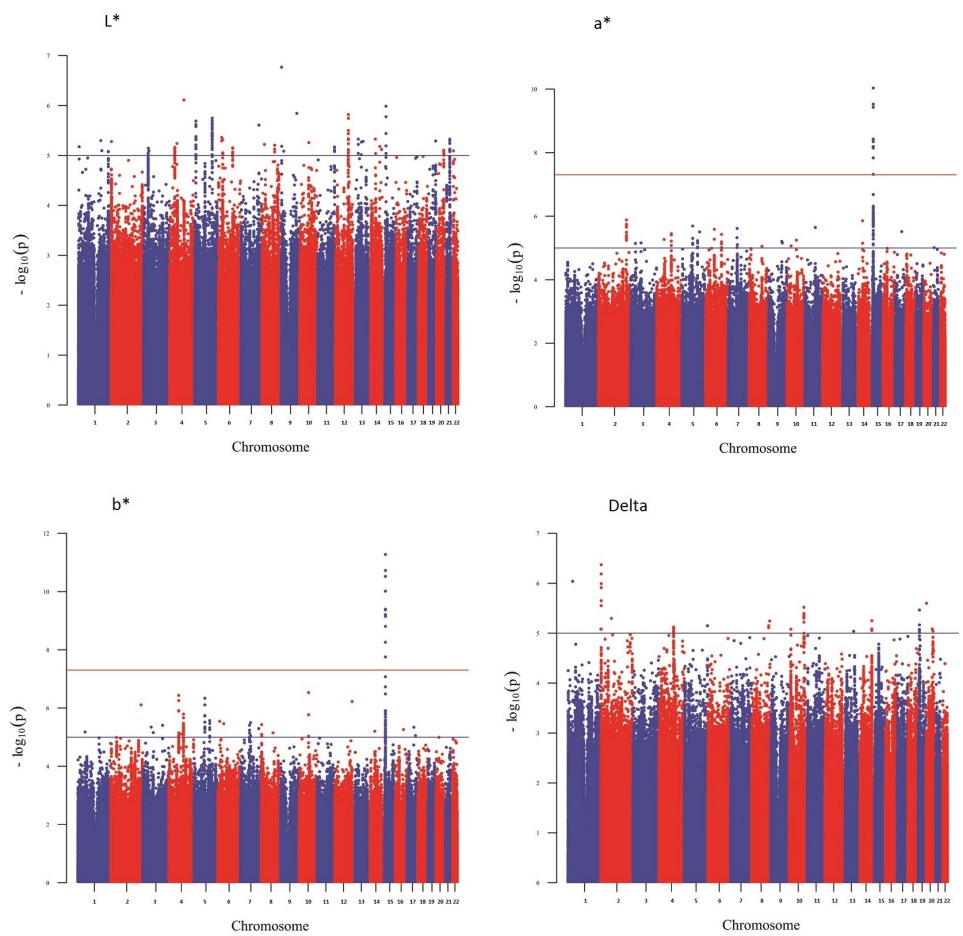

**Figure 2** Manhattan plot reporting the association results for iris color (L⋆, a⋆, b⋆ and delta).

allele was significantly associated with lighter skin pigmentation ($p = 0.003$, Bonferroni-corrected significance threshold $p = 0.01$). This marker also had a concordant effect on the College student sample (A allele associated with lighter skin pigmentation—higher L⋆ values), but the $p$-values were not nominally significant. In the Taizhou cohort, in which melanin values were measured in the upper arm, rs2373391 was also not significant and the regression coefficient was close to 0. None of the remaining markers was nominally significant in any of the three replication samples.

## Association results for loci identified in European populations

We explored in our East Asian sample the allele frequencies and effect sizes of loci identified in European populations. These results are depicted in Table S1. The table includes two non-synonymous variants in the genes *SLC24A5* (rs1426654) and *SLC45A2* (rs16891982), which have derived alleles with very high frequencies in Europe and very strong effects on skin pigmentation in admixed groups (*Lamason et al., 2005*; *Norton et al., 2007*). The table also includes variants that reached genome-wide significance in the largest GWAS of skin pigmentation in

**Table 2  Main signals observed in the GWAS of iris color (L*, a*, b* and delta).** For each trait, we report the *P*-values obtained for all the other traits in the association study, as well as the log10 Bayes Factor obtained in the joint analysis of L*,a* and b*.

| | CHR | POS | NEA/EA | *Gene* | Frq EA | INFO | Beta | SE | P(L*) | P(a*) | P(b*) | P(delta) | BF |
|---|---|---|---|---|---|---|---|---|---|---|---|---|---|
| **L*signals** | | | | | | | | | | | | | |
| rs6664080 | 1 | 226797426 | T/C | *C1orf95* | 0.297 | 0.993 | 0.362 | 0.08 | 8.36E–06 | 2.56E–03 | 2.88E–02 | 3.42E–03 | 2.774 |
| rs4425211 | 3 | 36951898 | A/G | *TRANK1* | 0.072 | 0.959 | −0.676 | 0.148 | 7.14E–06 | 6.79E–05 | 6.33E–04 | 5.85E–01 | 1.678 |
| rs12501370 | 4 | 41043870 | G/C | *APBB2* | 0.556 | 0.982 | −0.332 | 0.073 | 6.89E–06 | 3.14E–04 | 1.93E–03 | 7.97E–01 | 2.501 |
| rs2658084 | 5 | 10136135 | C/T | *CTD-2199O4.1* | 0.835 | 0.97 | −0.502 | 0.104 | 2.05E–06 | 4.11E–03 | 3.32E–02 | 2.29E–01 | 2.976 |
| rs113633047 | 5 | 133402895 | G/T | | 0.282 | 0.95 | 0.419 | 0.087 | 2.00E–06 | 2.25E–04 | 5.94E–04 | 1.09E–01 | 2.735 |
| rs6924266 | 6 | 25505617 | C/T | *LRRC16A* | 0.233 | 0.98 | −0.408 | 0.087 | 4.38E–06 | 3.64E–05 | 1.05E–04 | 8.77E–02 | 2.583 |
| NA | 6 | 32626040 | C/T | | 0.075 | 0.996 | −0.64 | 0.138 | 4.81E–06 | 6.29E–05 | 1.79E–03 | 2.07E–01 | 2.003 |
| rs9373973 | 6 | 108079595 | T/A | *SCML4* | 0.185 | 0.938 | −0.454 | 0.099 | 7.02E–06 | 2.19E–03 | 1.12E–02 | 9.11E–02 | 2.36 |
| rs145048184 | 8 | 97070341 | C/T | | 0.084 | 0.962 | 0.643 | 0.14 | 6.23E–06 | 8.89E–06 | 3.05E–05 | 5.74E–01 | 1.947 |
| rs610106 | 11 | 128935028 | T/C | *ARHGAP32* | 0.814 | 0.97 | 0.44 | 0.096 | 6.79E–06 | 9.91E–04 | 5.40E–03 | 9.39E–01 | 2.318 |
| rs249625 | 12 | 98185248 | A/G | | 0.584 | 0.979 | −0.364 | 0.077 | 3.16E–06 | 5.72E–03 | 3.77E–03 | 9.27E–03 | 3.352 |
| rs76930569 | 15 | 28196145 | C/T | *OCA2* | 0.623 | 1 | 0.38 | 0.076 | 1.03E–06 | **9.27E–11** | **5.30E–12** | 3.42E–01 | 7.007 |
| rs75161997 | 20 | 55701691 | C/T | | 0.06 | 0.992 | −0.732 | 0.161 | 7.84E–06 | 7.49E–04 | 1.77E–03 | 8.34E–01 | 1.485 |
| rs8131065 | 21 | 38011676 | C/T | | 0.215 | 0.979 | −0.419 | 0.09 | 4.75E–06 | 1.49E–02 | 1.25E–01 | 4.43E–02 | 3.478 |
| **a*signals** | | | | | | | | | | | | | |
| rs55821297 | 4 | 111399598 | A/G | *ENPEP* | 0.771 | 0.926 | 0.435 | 0.092 | 2.96E–04 | 3.55E–06 | 2.19E–06 | 4.86E–05 | 2.710 |
| rs72763726 | 5 | 82044846 | C/T | | 0.090 | 0.950 | 0.652 | 0.135 | 1.32E–04 | 2.04E–06 | 4.60E–07 | 7.57E–01 | 2.349 |
| rs330203 | 5 | 119016414 | C/G | *CTC-507E12.1* | 0.500 | 0.996 | −0.343 | 0.075 | 4.05E–03 | 7.02E–06 | 3.40E–06 | 8.74E–01 | 2.997 |
| rs9345521 | 6 | 65511281 | C/A | *EYS* | 0.467 | 0.976 | 0.352 | 0.074 | 1.64E–04 | 2.59E–06 | 6.92E–05 | 5.27E–01 | 2.879 |
| rs78001527 | 6 | 120168240 | G/C | | 0.074 | 0.973 | −0.686 | 0.146 | 1.65E–03 | 3.74E–06 | 6.68E–05 | 9.24E–01 | 1.892 |
| rs6977845 | 7 | 67216370 | T/A | | 0.093 | 0.881 | 0.666 | 0.139 | 2.58E–04 | 2.42E–06 | 5.04E–05 | 7.82E–01 | 1.979 |
| rs76930569 | 15 | 28196145 | C/T | *OCA2* | 0.623 | 1.000 | 0.497 | 0.074 | 1.03E–06 | **9.27E–11** | **5.30E–12** | 3.42E–01 | 7.007 |
| **b*signals** | | | | | | | | | | | | | |
| rs12510870 | 4 | 74358277 | T/C | *AFM* | 0.190 | 0.994 | −0.496 | 0.096 | 1.06E–01 | 3.37E–04 | 3.68E–07 | 1.60E–01 | 4.614 |
| rs1996603 | 4 | 111378362 | G/A | *ENPEP* | 0.779 | 0.960 | 0.451 | 0.092 | 1.32E–03 | 1.22E–05 | 1.61E–06 | 1.60E–03 | 2.728 |
| rs72763726 | 5 | 82044846 | C/T | | 0.090 | 0.950 | 0.691 | 0.134 | 1.32E–04 | 2.04E–06 | 4.60E–07 | 7.57E–01 | 2.349 |
| rs330203 | 5 | 119016414 | C/G | *CTC-507E12.1* | 0.500 | 0.996 | −0.355 | 0.075 | 4.05E–03 | 7.02E–06 | 3.40E–06 | 8.74E–01 | 2.997 |
| rs141034411 | 8 | 3098640 | C/A | *CSMD1* | 0.413 | 0.935 | 0.353 | 0.075 | 9.47E–05 | 3.20E–05 | 3.69E–06 | 5.71E–01 | 2.843 |
| rs2278745 | 10 | 71152091 | T/C | *HK1* | 0.602 | 0.966 | −0.395 | 0.075 | 3.05E–03 | 5.70E–06 | 2.95E–07 | 1.51E–01 | 3.668 |
| rs76930569 | 15 | 28196145 | C/T | *OCA2* | 0.623 | 1.000 | 0.527 | 0.074 | 1.03E–06 | **9.27E–11** | **5.30E–12** | 3.42E–01 | 7.007 |
| **Delta signals** | | | | | | | | | | | | | |
| rs72776813 | 2 | 2792075 | G/A | | 0.211 | 0.980 | −0.481 | 0.093 | 1.18E–03 | 8.21E–02 | 7.15E–02 | 4.25E–07 | 1.226 |
| rs243946 | 4 | 111300362 | C/T | *ENPEP* | 0.481 | 0.984 | −0.339 | 0.075 | 1.52E–04 | 3.60E–03 | 1.99E–03 | 7.74E–06 | 1.716 |
| rs16904127 | 8 | 130628606 | G/A | *CCDC26* | 0.469 | 0.904 | −0.359 | 0.079 | 2.30E–01 | 8.01E–01 | 5.78E–01 | 7.13E–06 | 0.132 |
| rs7914735 | 10 | 110441149 | C/T | | 0.537 | 0.988 | −0.356 | 0.075 | 6.46E–02 | 3.40E–01 | 3.54E–01 | 3.03E–06 | −0.227 |
| NA | 14 | 106875131 | A/G | | 0.522 | 0.884 | −0.374 | 0.081 | | | | 5.63E–06 | |
| rs11667379 | 19 | 9126468 | G/C | | 0.411 | 0.954 | 0.348 | 0.074 | 1.70E–02 | 3.44E–01 | 2.92E–01 | 3.45E–06 | 0.532 |

Rawofi et al. (2017), *PeerJ*, DOI 10.7717/peerj.3951

**Table 3  Results of the replication effort for skin pigmentation signals in three independent Chinese samples.** Note that in the Physical Examination and Taizhou cohorts melanin was reported as M-index (higher *M*-values indicate darker skin), whereas in the sample of College students melanin was reported as L* (higher L* values indicate lighter skin).

| SNP | CHR | POS | NEA/EA | This study | | Physical examination cohort | | Taizhou cohort | | College Students | | | |
| --- | --- | --- | --- | --- | --- | --- | --- | --- | --- | --- | --- | --- | --- |
| Site | | | | Upper arm | | Buttocks | | Upper arm | | Buttocks | | Upper arm | |
| Index | | | | M-index | | M-index | | M-index | | L* | | L* | |
| | | | | *Beta* | *P*-value | *Beta* | *P*-value | *Beta* | *P*-value | *Beta* | *P*-value | *Beta* | *P*-value |
| rs2003589 | 2 | 217527465 | T/C | 2.464 | 1.05E–06 | −0.104 | 0.864 | 0.171 | 0.575 | −1.927 | 0.259 | −0.659 | 0.635 |
| rs853807 | 5 | 67752638 | C/T | −0.961 | 9.75E–06 | −0.243 | 0.440 | −0.050 | 0.755 | 0.085 | 0.744 | −0.018 | 0.931 |
| rs57836066 | 6 | 71304950 | T/G | −1.838 | 2.40E–06 | 0.605 | 0.225 | 0.213 | 0.394 | 0.995 | 0.317 | 0.175 | 0.828 |
| rs2373391 | 7 | 88449300 | T/A | −1.076 | 5.09E–06 | **−1.132** | **0.003** | 0.024 | 0.889 | 0.120 | 0.739 | 0.273 | 0.349 |
| rs7945369 | 11 | 103425586 | T/C | −1.159 | 3.97E–07 | −0.614 | 0.051 | 0.211 | 0.190 | NA | NA | NA | NA |

populations of European ancestry, which included more than 17,000 samples (*Liu et al., 2015*). Five of the six variants are either absent or present at very low frequencies in the East Asian sample and have relatively low imputation scores (info < 0.8). The only exception is the polymorphism rs4268748 located near the *MC1R* gene, in which the minor allele (allele C) has a frequency of 27.4% and a good imputation score (info = 0. 983). None of the variants reached nominal significance in our East Asian sample, except the *SLC24A5* rs1426654 non-synonymous variant, which is present at a very low frequency in the sample (frequency of the derived A allele 1.6%).

## DISCUSSION

In this paper, we describe the results of a genome-wide association study of pigmentary traits (skin pigmentation and iris color) in East Asian populations. We were able to confirm the important role that the gene *OCA2* plays in East Asian populations. In our iris color GWA, we observed a genome-wide significant signal in the *OCA2* region. The lead SNP in this region was rs76930569, and this marker showed particularly strong associations with the a* and b* color dimensions ($p$-values of $9.27 \times 10^{-11}$ and $5.30 \times 10^{-12}$, respectively). However, rs76930569 was not associated with delta, which is a measure that captures iris heterochromia (e.g., the color differences between the pupillary and ciliary regions of the iris). This indicates that the genetic architecture of iris color is different from the genetic architecture of iris heterochromia. It is important to note that rs76930569 is in very strong linkage disequilibrium with the non-synonymous SNP rs1800414 ($R^2 = 1$ in CHB, CHS and JPT 1KG samples). Not surprisingly, the $p$-values of rs1800414 were very similar to those of rs76930569 in our sample, and most probably this non-synonymous SNP is the causal polymorphism driving the association. A recent study (*Yang et al., 2016*) using cultured melanocytes, and transgenic and targeted gene modification analyses on zebrafish and mouse have shown that the rs1800414 G variant (Arg615) is functional and leads to skin lightening. In our skin pigmentation analysis, the rs1800414 G variant was also associated with lighter skin pigmentation, but did not reach genome-wide significance (beta $= -0.81$, $p = 5.5 \times 10^{-4}$). The magnitude of the skin pigmentation effect observed in our study is quite similar to the effects described in previous reports in which the M-index was used to describe constitutive pigmentation (e.g., *Edwards et al., 2010*, beta $= -1.26$ in a sample of individuals of East Asian ancestry living in Canada, and beta $= -0.86$ in a Han Chinese sample; *Eaton et al., 2015*, beta $= -0.91$ in a sample of individuals of East Asian ancestry living in Canada).

Aside from the eye color signal in the *OCA2* region, no other regions surpassed the genome-wide significance threshold for skin pigmentation or iris color. However, there were several regions harboring multiple common markers with good imputation scores and suggestive $p$-values ($p < 10^{-5}$; Tables 1 and 2; File S1). For skin pigmentation, we followed up the lead SNPs for each region in three independent Chinese samples (Table 3). One of the SNPs, rs2373391, which is located in an intron of the gene *ZNF804B*, replicated in one of the Chinese samples (beta $= -1.132$, $p = 0.003$). The gene *ZNF804B* encodes a zinc finger protein, but it has been poorly characterized. Variants within this gene have been

nominally associated with a number of traits in previous GWA studies (Anorexia nervosa, *Wang et al., 2011*; Heschl's gyrus morphology, (*Cai et al., 2014*; and IgG glycosylation, *Lauc et al., 2013*, among others), but to our knowledge no associations with pigmentary traits have been reported for this or nearby genes. Given that we were able to replicate the association only in one of the three East Asian samples, it will be important to investigate the potential role of this region in skin pigmentation in larger samples from East Asia. None of the other regions showing suggestive associations with skin pigmentation were replicated in the follow-up samples.

Unfortunately, we could not follow up any of the suggestive regions identified for iris color, because of the lack of studies in East Asia reporting quantitative measures of iris color. In this study, we show that there is a substantial amount of variation in iris color in East Asia (Fig. S2), and that the *OCA2* region (and more particularly, the non-synonymous variant rs1800414) is a major determinant of the variation observed. We estimated that rs1800414 explains 11.9%, 10.4% and 6% of the variation observed in the b⋆, a⋆ and L⋆ coordinates in our sample, respectively. Therefore, there is a substantial amount of variation that remains unexplained in our sample. This highlights the need to carry out more studies using quantitative measures of iris color in East Asia. Categorical definitions of iris color would not be useful in East Asian populations, because in such a classification most of the irises would have been categorized as "brown", but there is a substantial range of L⋆, a⋆ and b⋆ values within this category.

To our knowledge, this is the first genome-wide association study of pigmentary traits in East Asian populations. The main weakness of this study is the small sample size, which limits our statistical power to identify variants with small effects on skin and iris pigmentation. Another limitation is that constitutive pigmentation was measured in different body sites (e.g., buttocks or inner arm) in the discovery and replication samples. Ideally, constitutive pigmentation should be measured in the buttocks, because there is less exposure to UV in this body site. However, often it is not possible to obtain pigmentation measures in the buttocks, and constitutive pigmentation is measured in the inner upper arm. In this site, there may be some exposure to UV light, particularly at certain times of the year, and several studies have reported that measurements of pigmentation in the inner arm are not interchangeable with estimates in the buttocks (*Johansen et al., 2016*; *Bieliauskiene, Philipsen & Wulf, 2017*). In the discovery sample from Canada, inner upper arm measures were collected either in the winter/early spring (January to April) or the fall (October and November), so it would not be expected that UVR exposure would have a substantial effect on melanin levels. Similarly, in the two replication samples in which pigmentation measures were taken in the inner arm, Taizhou cohort and College students, the measures were collected in April, and September, respectively. Additionally, it is important to note that in one of the replication samples, pigmentation was reported as the CIELAB L⋆ value, instead of the M-index. Although these two measures are highly correlated (*Shriver & Parra, 2000*), ideally all the discovery and replication samples should have been measured using the same index. Given that measurements were taken in different body sites, and for one replication sample using different pigmentation units, we could not carry out a meta-analysis of the discovery and replication samples. Instead, in our
replication effort, we evaluated if the SNPs that were followed up in the replication samples reached significance after Bonferroni-correction based on the number of independent tests. Finally, it is important to note that for this GWAS we used both directly genotyped and imputed data. This is the standard protocol used in this type of studies, because it increases the power to identify associated variants (*Spencer et al., 2009*; *Marchini & Howie, 2010*). We implemented two different strategies to minimize potential problems related to imputation: (1) we used tests implemented in the program SNPTEST that take into account uncertainty in the imputed genotypes, and (2) we only followed up markers with very high imputation scores (info > 0.8).

In spite of these limitations, this study provides important insights about the genetic architecture of skin and iris color in East Asian populations. Using the program Quanto (http://biostats.usc.edu/Quanto.html) we estimated that our skin pigmentation association study ($N = 305$) had good power (>0.8) to identify variants explaining approximately 12.5% of the variance of the trait. This implies that there are no polymorphic variants in our samples with large effects on skin melanin levels. For iris color ($N = 342$), our study had good power to identify variants explaining approximately 11% of the variance of the trait, and in fact we identified a SNP with an effect of approximately that size. It will be critical to expand the number of genome-wide association studies in East Asian populations in order to be able to identify genetic markers with smaller effects on skin and iris pigmentation. We cannot exclude the possibility that there are variants with strong effects on pigmentation that have been fixed in East Asian populations due to the action of positive selection. This is something that happened in Europe with the well-known *SLC24A5* rs1426654 A allele, which is the variant with the largest effect on melanin levels reported in human populations (*Lamason et al., 2005*). Identifying this type of variants, if they exist in East Asian populations, would require association studies in admixed samples with a substantial East Asian contribution.

We followed up in our East Asian sample genetic markers in 6 loci that have been reported to influence skin pigmentation in European populations (*SLC45A5, IRF4, HERC2, SLC24A5, DEF8/MC1R* and *RALY/ASIP,* Table S1). Most of these variants are absent (*RALY/ASIP* rs6059655) or present at very low frequencies in the East Asian sample (*SLC45A2* rs16891982, *IRF4* rs12203592, *HERC2* rs12913832 and *SLC24A5* rs4268748, Table S1). Because of their low frequencies, these SNPs explain little variation of skin pigmentation in East Asian populations. However, it is important to note that the estimated effect size of the derived *SLC24A5* rs1426654 A allele in our GWAS is quite large (beta = −2.46), in agreement with numerous reports indicating that this is the locus with the strongest effect on skin pigmentation so far described in human populations (*Lamason et al., 2005*; *Basu Mallick et al., 2013*; *Beleza et al., 2013a*; *Beleza et al., 2013b*). Only one of the variants reaching genome-wide significance in a large GWAS study in a European sample (*DEF8/MC1R* rs4268748; *Liu et al., 2015*) is present with relatively high frequencies in our East Asian sample. In fact, the allele reported to decrease pigmentation in the European sample (C allele) is slightly more frequent in East Asian populations than in European populations (28.5% in EAS 1000 Genomes Project sample vs. 22.8% in EUR 1000 Genomes project). However, the rs4268748 polymorphism is not nominally significant in our sample.

One potential explanation is that rs4268748 is in linkage disequilibrium (LD) with causal loci in Europe, and the pattern of LD between rs4268748 and the causal loci, or the allele frequencies of the causal loci are different in European and East Asian populations. Supporting this explanation is the detailed analysis of the *MC1R* region reported for one of the European samples (Rotterdam study) included in *Liu et al. (2015)* GWAS. These authors described that 3 known high-penetrance variants located within the *MC1R* gene (rs1805007 (R151C), rs1805008 (R160W), rs1805009 (D294H)) showed the most significant association with skin pigmentation in the Rotterdam study sample, and replacing the SNP rs4268748 by the high penetrance variants marginally increased the amount of the phenotypic variance explained. These three *MC1R* high-penetrance variants, two of which have frequencies higher than 5% in Europe (rs1805007 and rs1805008), are not found or are found in extremely low frequencies in East Asia, and this may explain the absence of an association signal in our study. *Yamaguchi et al. (2012)* reported that a low-penetrance *MC1R* variant, rs885479 (R163Q), which is common in East Asia (frequency derived 163Q allele in EAS 1000 Genomes project sample = 61.6%) but much less frequent in Europe was nominally associated with light skin in a Japanese sample. As expected, the rs885479 polymorphism was present in our East Asian sample, with a frequency of the derived 163Q allele of 60.1%. However, this variant was not nominally associated with skin pigmentation in our sample (beta = $-0.046$, $p = 0.843$).

Overall, our study confirms previous research indicating that the evolution of light skin in East Asia and Europe took place, at least to a large extent, independently in both groups after the split of the ancestral East Asian and European populations following the Out-of-Africa migration. In Europe, functional derived variants in the genes *SLC24A5*, *SLC45A2* and *OCA2/HERC2* increased in frequency as a result of positive selection (*Lamason et al., 2005*; *Izagirre et al., 2006*; *Norton et al., 2007*; *Lao et al., 2007*; *Wilde et al., 2014*). In East Asia, the non-synonymous polymorphism rs1800414 located within the *OCA2* gene dramatically increased in frequency, also as a result of an independent selective event (*Lao et al., 2007*; *Edwards et al., 2010*; *Donnelly et al., 2012*; *Hider et al., 2013*). This constitutes, with lactase persistence and adaptation to altitude, one of the most fascinating examples of convergent evolution in human populations.

## CONCLUSION

Here we report the first genome-wide association of pigmentary traits (skin pigmentation and iris color) in East Asian populations. We measured these traits using quantitative methods, and we show that there is considerable variation not only in skin pigmentation but also in iris color, in spite of the fact that using categorical classifications, all the irises in this sample would have been categorized as "brown". We confirmed the important role that the *OCA2* gene plays in normal pigmentation variation in East Asian populations. In particular, the non-synonymous variant rs1800414 explains a substantial amount of variation in iris color. We did not observe any genome-wide significant variant for skin pigmentation, but one of the lead SNPs showing suggestive significance in our study, rs2373391, which is located in an intron of the ZNF804B gene on chromosome 7, was

replicated in an independent Chinese sample ($p = 0.003$). It will be critical to carry out additional association studies in East Asian populations in order to uncover additional variants with smaller effects on skin pigmentation and iris color.

## ACKNOWLEDGEMENTS

We would like to thank all the individuals who participated in this study.

### Funding

Lida Rawofi was funded by a Natural Sciences and Engineering Research Council (NSERC) CGS-M scholarship. Esteban J. Parra was funded by an NSERC Discovery Grant. Heather L. Norton and Esteban J. Parra were funded by the US National Institute of Justice (grant 2013-DN-BX-K011). In Canada, computations were performed on the GPC supercomputer at the SciNet HPC Consortium. SciNet is funded by: the Canada Foundation for Innovation under the auspices of Compute Canada; the Government of Ontario; Ontario Research Fund—Research Excellence; and the University of Toronto. Yanyun Ma, Jiucun Wang and Li Jin were funded by grants from the National Science Foundation of China (31521003) the Science and Technology Committee of Shanghai Municipality (16JC1400500) and the 111 project (B13016) from Ministry of Education of P.R.China. Bing Su was funded by grants from the Strategic Priority Research Program of the Chinese Academy of Sciences (XDB13010000) and the National Natural Science Foundation of China (91631306). The funders had no role in study design, data collection and analysis, decision to publish, or preparation of the manuscript.

### Grant Disclosures

The following grant information was disclosed by the authors:
Natural Sciences and Engineering Research Council (NSERC) CGS-M scholarship.
NSERC Discovery Grant.
US National Institute of Justice: 2013-DN-BX-K011.
Canada Foundation for Innovation.
Government of Ontario; Ontario Research Fund—Research Excellence.
University of Toronto.
National Science Foundation of China: 31521003.
Science and Technology Committee of Shanghai Municipality: 16JC1400500.
Ministry of Education of P.R. China: B13016.
Chinese Academy of Sciences: XDB13010000.
National Natural Science Foundation of China: 91631306.

### Competing Interests

The authors declare there are no competing interests.
## Author Contributions

- Lida Rawofi analyzed the data, wrote the paper, prepared figures and/or tables, reviewed drafts of the paper.
- Melissa Edwards performed the experiments, analyzed the data.
- S Krithika performed the experiments.
- Phuong Le and David Cha analyzed the data.
- Zhaohui Yang and Yanyun Ma performed the experiments, analyzed the data.
- Jiucun Wang and Bing Su analyzed the data, contributed reagents/materials/analysis tools, reviewed drafts of the paper.
- Li Jin analyzed the data, contributed reagents/materials/analysis tools.
- Heather L. Norton contributed reagents/materials/analysis tools, reviewed drafts of the paper.
- Esteban J. Parra conceived and designed the experiments, analyzed the data, contributed reagents/materials/analysis tools, wrote the paper, prepared figures and/or tables, reviewed drafts of the paper.

## Human Ethics

The following information was supplied relating to ethical approvals (i.e., approving body and any reference numbers):

The study was approved by the University of Toronto Research and Ethics Board (Protocol Reference #27015).

## Data Availability

The raw data has been provided as a Supplemental File.

## Supplemental Information

Supplemental information for this article can be found online at http://dx.doi.org/10.7717/peerj.3951#supplemental-information.

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
