# Peer review of "Genome-wide association study of pigmentary traits (skin and iris color) in individuals of East Asian ancestry"

_PeerJ, doi:10.7717/peerj.3951_

## Round 0.1 · original submission · Minor Revisions

Address concerns of both reviews with priority on the following:

1. In the discussion include description of skin tone measurements made from arm versus buttock (see Reviewer 1's comments for more details).
2. In the Discussion describe limitations and potential issues with the imputation.
3. Include more description on the follow-up sample collections including replicates.
4. Consider including a table with previous pigmentation loci from other other populations with the Beta-estimate.
5. Ensure that the quality of figures is improved. (See reviewer 1 comments).
6. Include a combined analysis of the discovery and replication set.
6. Consider the addition of more publications.

·

Basic reporting

This is a important piece of work and well investigated. I have only minor points that should be adressed. Unfortunately due to the relatively small sample size no, so far unknown variants, showed significant association with eye and skin colour. I would, however encourage the authors to investiage ZNF804B in more details and in larger datasets if possible.The level of the English throughout the manuscript is very high and, therefore the manuscript is easy to read and well structured. The figures and tables in the manus are overall of high scientific quality and support the content of the manus nicely, however the graphic overall quality of the figures is low and eg. the abscissa in Figure 1 and 2 is inadequate.
The use of literature is satisfying, but should be supplemented with more literature. Lines 48-52: Andersen et al. 2013 (Forensic Sci Int Genet. 2013 Sep;7(5):508-15. doi: 10.1016/j.fsigen.2013.05.003).

Experimental design

I like the overall study design with a study population and follow up population to replicate the skin pigmentation findings. The statistical analyses are wisely choosen and performed in a very nice way.
I see some challenges with the measurements of skin pigmentation and the use of imputatation. The authors have obtained buttock pigmentation measurements for the study population, but not for all of the follow up populations. It is well known that buttock levels resemble the constitutive pigmentation levels, whereas arm levels may be influenced by environmental factors (UVR) (eg. Johansen et al. 2013, PLoS One. 2016 Mar 3;11(3):e0150381. doi: 10.1371/journal.pone.0150381). I find it problematic to compare these two areas of measurements without any further discussion. This should be adressed in the discussion section. Also a better discription upon the sample collection for the follow-up samples is needed. Much could be learned from the very nice description of the study population "Measurement of pigmentary traits". Eg. how many replicates were taken and did the authors use the median of the replicates if any?
I would also appreciate a discussion about the use of imputation.

Validity of the findings

The outsome of the study is important for research within the field of pigmentation genetics. The East Asians is a neclegted population groups. It is not surprising that no variants show significant association with skin colour as the variation within the investigated group is minor. A the authors also describes, the number of samples needs to be increased in order to find variants with minor effects.

Reviewer 2 ·

Basic reporting

no comment

Experimental design

no comment

Validity of the findings

no comment

Additional comments

This manuscript reported a GWAS on pigmentation in Asian populations. Such study is much needed. The strength of this study included instrument-based pigmentation phenotype measurement. The reviewer only has two suggestive comments.
1. The authors did not report meta analysis of the discovery set and replication set. Any reason for it?
2. It would be helpful to list the results of all the pigmentation loci previously identified in white populations. It is important to compare the beta estimates between the two racial groups. Of course this study has a much smaller sample size so that P value is not comparable.

---

## Round 0.2 · accepted · Accept

Thank you for addressing all the points raised by the reviewers.